# A Novel Deep Learning Architecture with Multi-Scale Guided Learning for Image Splicing Localization

Zhongwang Li [1], Qi You [2] and Jun Sun [2,*]

1    School of Electrical Engineering, Wuhu Institute of Technology, Wuhu 241006, China; lizhongwang@whit.edu.cn
2    School of Artificial Intelligence and Computer Science, Jiangnan University, Lihu Avenue, Wuxi 214122, China; 7171905005@stu.jiangnan.edu.cn
*    Correspondence: junsun@jiangnan.edu.cn

**Abstract:** The goal of image splicing localization is to detect the tampered area in an input image. Deep learning models have shown good performance in such a task, but are generally unable to detect the boundaries of the tampered area well. In this paper, we propose a novel deep learning model for image splicing localization that not only considers local image features, but also extracts global information of images by using a multi-scale guided learning strategy. In addition, the model integrates spatial and channel self-attention mechanisms to focus on extracting important features instead of restraining unimportant or noisy features. The proposed model is trained on the CASIA v2.0 dataset, and its performance is tested on the CASIA v1.0, Columbia Uncompressed, and DSO-1 datasets. Experimental results show that, with the help of the multi-scale guided learning strategy and self-attention mechanisms, the proposed model can locate the tampered area more effectively than the state-of-the-art models.

**Keywords:** image splicing localization; multi-scale guided learning; image forensics; self-attention mechanism

## 1. Introduction

With the development of image-editing tools, image forgery has become so easy and low-cost that it is difficult for the human eye to distinguish traces of image forgery. The most common methods of image tampering include image splicing, copy–move and removal, and examples of these manipulations, as shown in Figure 1. The detection of tampered images by splicing operation, as the focus of this work, is a procedure of copying the selected regions from one image and then pasting them into another image. Previous splicing detection methods mainly emphasize the characteristics of image processing. Some examples of such types of methods are Color Filter Array (CFA) [1], Photo Response Non-Uniformity (PRNU) [2], and image residual analysis [3], to name only a few. Although these methods are reliable for uncompressed images, they may fail to detect the tampered areas of compressed images; moreover, it is time-consuming for them to extract handcrafted features.

Convolutional neural networks (CNNs) have achieved excellent performance in various computer vision tasks, such as object detection [4,5] and semantic segmentation [6,7]. Recently, CNNs have also been applied to image tampering detection and have shown good performance in this task. Compared to traditional image processing methods, CNN-based methods can automatically construct complex image features (i.e., handcrafted features) during image processing. However, many CNN-based image splicing localization methods [8–10] do not take global image information into consideration, making the models prone to disturbance by the semantic information of the image content.

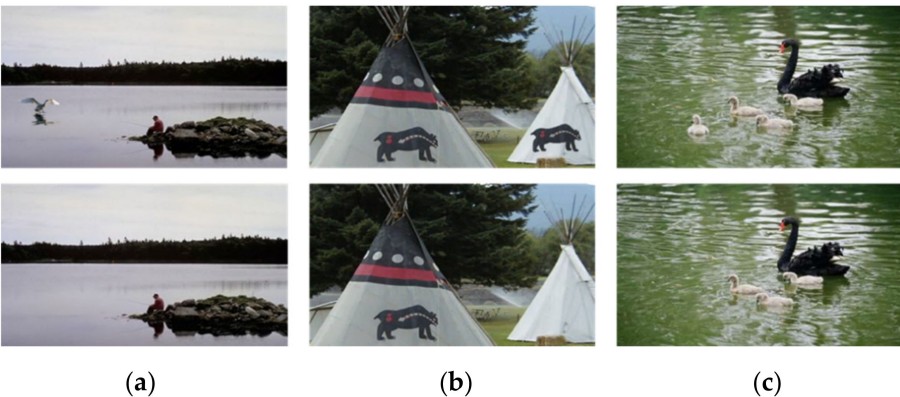

<div align="center">(<b>a</b>)        (<b>b</b>)        (<b>c</b>)</div>

**Figure 1.** The figure demonstrates three types of image manipulation, namely (**a**) splicing, (**b**) copy–move, and (**c**) removal. The tampered images are on the top, and the authentic image are on the bottom.

In this paper, we propose a novel deep learning architecture with a fully convolutional network (FCN) to localize the tampered area of an image. This architecture incorporates the multi-scale guided learning strategy and self-attention mechanisms. The multi-scale guided learning strategy is employed to scale the tampering mask with different sizes, and then calculate the binary classification loss in each feature extraction layer and feature fusion layer. Our architecture uses self-attention mechanisms, including spatial and channel self-attention mechanisms, for the purpose of promoting that the model pay more attention to important features while restraining unimportant or noisy features. Our proposed architecture adopts ResNet-50 [11] as the backbone, since it can extract more complex and effective image features with fewer parameters compared to those of VGG-16 [12].

The rest of this paper is organized as follows: in Section 2, we briefly review the related work; in Section 3, we describe the proposed model in detail. Section 4 provides summaries of data preparation and implementation details, as well as the experimental results obtained by compared methods. We conclude this work in Section 5.

## 2. Related Work

The image forensics community has a long history, throughout which many image forgery detection algorithms have been developed. Based on the feature extraction method, we can roughly divide exiting image splicing localization algorithms into two classes, namely, traditional signal processing methods and deep learning methods, both of which we briefly discuss below.

### 2.1. Signal Processing Methods

Earlier image forgery detection algorithms are mainly based on image processing characteristics. In [13], the authors proposed a fine-grained analysis algorithm based on CFA interpolation patterns, assuming that image splicing may cause CFA interpolation discontinuities. Moreover, some researchers have used photo response non-uniformity (PRNU) [14] or scale invariant feature transform (SIFT) [15] to localize the tampered area. Although these algorithms are reliable for uncompressed images, they may fail to detect compressed images. To solve this issue, some researchers [16] have exploited traces left by JPEG compression to detect tampered areas, but this detection strategy may still fail on other formats of compressed images. In summary, the signal processing methods can only be applied under some specific circumstances.

### 2.2. Deep Learning Methods

Most splicing detection algorithms [17–19] based on CNNs can only deduce whether a given image is tampered with but cannot localize the tampered area. Zhang et al. [20] made a preliminary attempt to locate the tampering area, but their method can only detect

some rough square areas. Bappy et al. [8] proposed a hybrid CNN-LSTM model to capture the discriminative features in the boundaries between authentic and tampered regions, but the model fails to recognize image blocks that are completely within the tampered area. Salloum et al. [9] designed an FCN-based model that utilizes two output branches for multi-task learning, one of which is employed to localize the tampered area, and the second of which is used to localize the edge or boundary of the tampered area. Zhou et al. constructed a two-stream Faster R-CNN network [10], where RGB stream is used to extract features from the image and noise stream to discover the noise inconsistency between authentic and tampered regions. Moreover, there also exist some tamper detection methods [21,22] based on image similarity and semantic segmentation. However, the above deep learning methods ignore the importance of global information and are prone to disturbance by the semantic information of the image content.

## 3. Proposed Methods

In this section, we describe the details of our proposed network architecture, including the method of fusing multi-scale features and implementing self-attention mechanisms. This network architecture framework is shown in Figure 2. The rest of this section will illustrate our method in detail.

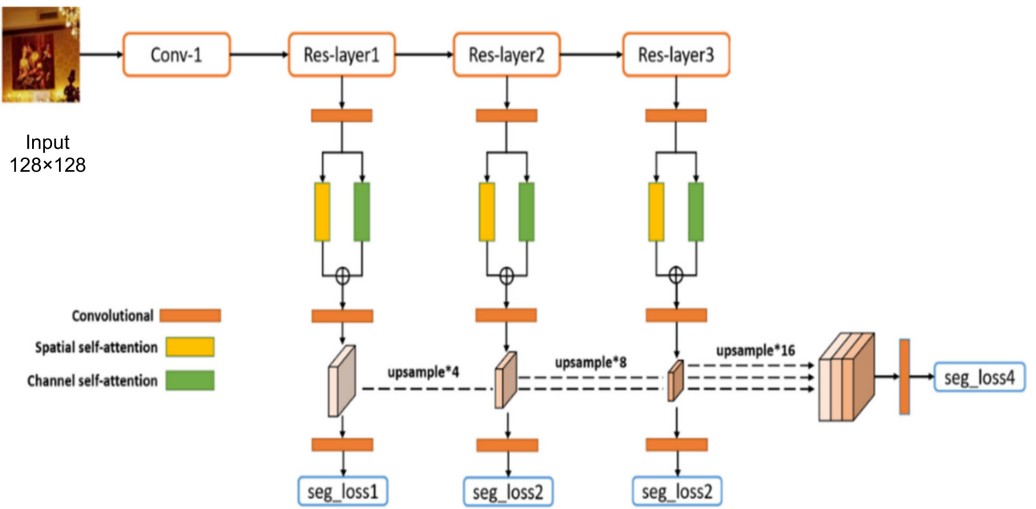

**Figure 2.** The proposed network architecture. The input is an image with the size 128 × 128, and the network outputs a tampered possibility map with the same size.

### 3.1. Network Architecture

Image splicing localization is similar to semantic segmentation, also making deep image semantic features useful for this task. For this reason, the proposed model adopts Resnet-50 as the backbone of the network to extract image features in each feature extraction layer (i.e., Res-layer1, Res layer2, and Res-layer3). At the beginning, a 1 × 1 convolutional layer is used to reduce the channels of feature maps, and then spatial and channel self-attention is integrated into the network to make the network pay more attention to important features in the spatial and channel dimensions. After that, an upsampling layer is employed to resize multi-scale feature maps to the size of the input image and concatenate them together. Finally, binary classification loss, including the seg_loss1, seg_loss2, seg_loss3, and seg_loss4, is calculated for training the model.

### 3.2. Multi-Scale Guided Learning

Unlike the method proposed in [9], which only calculates the loss-of-feature fused layer, our approach scales the tampering mask to the same size as those of the feature maps in each feature extraction layer, and generates a tampered probability map with each of its components ranging from 0 to 1. Here, 0 means authentic labeled pixel, and 1 means

tampered labeled pixel. We regard the pixels with tampered probability higher than $\eta$ as positive samples and the pixels with tampered probability lower than $\mu$ as negative samples. Otherwise, if a pixel is marked between $\mu$ and $\eta$, it is considered to be of an intermediate value. Since regarding the pixels with an intermediate value as either positive or negative samples may cause confusion, we ignore such pixels in the proposed method.

As such, each pixel in each feature extraction layer with respect to pixel label can be calculated as:

$$
L_{seg} = \begin{cases} \alpha \cdot P\left(x_i; F'_j\right)^{\gamma} \cdot \log\left(1 - P\left(x_i; F'_j\right)\right) & if\ y_i < \mu \\ \beta \cdot \left(1 - P\left(x_i; F'_j\right)^{\gamma}\right) \cdot \log P\left(x_i; F'_j\right) & if\ y_i < \eta \\ 0 & otherwise \end{cases} \tag{1}
$$

with

$$
\alpha = \frac{|Y^+|}{|Y^+| + |Y^-|}, \beta = \frac{|Y^-|}{|Y^+| + |Y^-|} \tag{2}
$$

where $Y^+$ and $Y^-$ denote the numbers of positive samples and negative samples in each training batch, respectively. Therefore, Equation (1) is very different from focal loss [23] in that $\alpha$ and $\beta$ in Equation (1) change adaptively, according to the proportion of positive and negative samples in each training batch. The focusing parameter $\gamma$ is used to smoothly adjust the rate of the decreasing weight of easy examples. The activation value and ground truth tampered probability of pixel $i$ are represented by $x_i$ and $y_i$, respectively; $P(x_i)$ is the standard sigmoid function; and $F'_j$ denotes different feature extraction layers.

With the above specification, the segmentation loss function for our model can be formulated as

$$
L_{seg\_total} = \sum_{j=1}^{J} \sum_{i=1}^{|I_j|} L_{seg}\left(x_i^j; F'_j\right) + \sum_{i=1}^{|I_{fuse}|} L_{seg}\left(x_i^{fuse}; F'_{fuse}\right) \tag{3}
$$

where $x_i^j$ is the activation value in each feature extraction layer $F'_j$, $x_i^{fuse}$ is the value from the feature fused layer $F'_{fuse}$, $|I_j|$ is the number of pixels in each feature extraction layer $F'_j$, and $\left|I_{fuse}\right|$ is the number of pixels in feature fusion layer $F'_{fuse}$.

By accumulating the binary classification loss of multi-scale feature maps, we can successfully activate the responses of different resolution feature maps to the corresponding resolution tampering regions, and weaken the effect of image content information. Results of ablation experiments show that this method can effectively accelerate the convergence speed and obtain better segmentation results.

### 3.3. Self-Attention Mechanisms

The convolution neural network in the proposed model can construct new feature maps from input feature maps, with the help of convolution kernels. After integrating self-attention mechanisms, the network emphasizes important features in the spatial and channel dimensions and discards those unimportant or noisy features. The spatial and channel self-attention mechanisms we use in our network are the same as the ones in CBAM [24]. The features that the spatial self-attention and the channel self-attention obtained are fused with a simple addition operation, as will be described below.

**Channel Self-attention.** The input feature $F$ obtained by the former convolutional layer first passes through a channel attention module to obtain the weighted $M_c(F)$ by the following formula

$$
M_c(F) = F \cdot M_c + MLP(AvgPool(F)) \tag{4}
$$

with

$$
M_c = \sigma(MLP(MaxPool(F))) \tag{5}
$$

where $F$ is the input feature map, $MaxPool$ and $AvgPool$ are the global maximum pooling layer and the global average pooling layer, respectively; $MLP$ is a multilayer perceptron;

$\sigma$ is the sigmoid activation function; and $M_c$ is the weight of channel features. Here, the fusion method is element-wise multiplication.

**Spatial Self-attention**. The input feature $F$ obtained by the former convolutional layer passes through a spatial attention module to obtain the weights $M_s(F)$. The calculation of $M_s(F)$ is as below

$$M_s(F) = F \cdot M_s \tag{6}$$

with

$$M_s = \sigma(f^{7 \times 7}([MaxPool(F); AvgPool(F)]) \tag{7}$$

where $F$ is the input feature map, *MaxPool* and *AvgPool* are the global maximum pooling layer and the global average pooling layer, respectively; $f^{7 \times 7}$ is a 7 × 7 size convolution kernel; $\sigma$ is the sigmoid activation function; and $M_s$ is the weight of spatial features. The fusion method is also element-wise multiplication.

## 4. Experiments

In this section, we illustrate the operation of dataset processing and the details of the training process. Furthermore, we provide the experimental results from our method on three common image manipulation datasets and compare them with those obtained by state-of-the-art algorithms on the same datasets.

### 4.1. Data Preparation

We trained our model using CASIA v2.0 dataset, which contains 7491 authenticated images and 5123 tampered images. The producers of the dataset did not save the mask of the ground truth but encoded the name of the tampered picture in the name of source pictures. Therefore, we used a simple and effective method to obtain the tampered area mask as

$$I_D = ||I_T - I_S|| \tag{8}$$

where $I_T$ and $I_S$ are the tampered image and source image, $I_D$ is the absolute difference of the gray values between $I_T$ and $I_S$. Here, 0 means that the pixel is labeled as "authentic", and 1 means that the pixel is labeled as "tampered". For each pixel, we took the pixels with $I_D$ higher than s for positive samples and the pixels with $I_D$ lower than s for negative samples. The formula is given as below

$$I_{truth} = \begin{cases} 1 & ||I_T - I_S|| > s \\ 0 & otherwise \end{cases} \tag{9}$$

We discarded the image generated by symmetry, and some images in which the tampered region boundary could not be approximately satisfied by the algorithm. After screening, we only used 4465 tampered images. However, the number of tampered image blocks was still not sufficient, so we performed a horizontal flip operation on the tampered images.

### 4.2. Parameters Setting

Training of the networks was performed in Pytorch using stochastic gradient descent (SGD) algorithm with a momentum of 0.9 and a weight decay of 0.0005, and the learning rate was initially set to 0.003 and was multiplied by 0.95 after every training epoch. We initialized the weights or parameters in our model with the weights pre-trained on the ImageNet dataset. We firstly trained the model with $L_{seg\_total}$, and then fine-tuned the trained model with hard negative examples.

### 4.3. Performance Evaluation Metrics

We evaluated the performance of our model and some existing methods by using the $F_1$ and Matthews Correlation Coefficient (*MCC*) metrics, both of which are per-pixel localization metrics.

Both $F_1$ and *MCC* metrics require a binary mask as input. We converted each output into a binary mask based on a threshold (equal to 0.5 here), and then calculated the $F_1$-socre and *MCC* indicators by comparing the output binary mask with the corresponding ground truth mask. The $F_1$ score is defined as:

$$F_1(M_{out}, M_{gt}) = \frac{2TP}{2TP + FN + FP} \tag{10}$$

where $M_{out}$ represents the binary mask of output and $M_{gt}$ denotes the ground truth mask. *TP* is known as the number of pixels which are correctly classified as spliced, *TN* as the number of pixels which are correctly classified as authentic, *FN* as the number of pixels which are incorrectly classified as authentic, and *FP* as the number of pixels which are incorrectly classified as spliced. The *MCC* metric is defined as:

$$MCC(M_{out}, M_{gt}) = \frac{TP \times TN - FP \times FN}{\sqrt{(TP + FP)(TP + FN)(TN + FP)(TN + FN)}} \tag{11}$$

*4.4. Experimental Results*

In this section, four image datasets were totally tested. They are CASIA v1.0, CASIA v2.0, and Columbia Uncompressed and DSO-1 datasets, which are summarized in Table 1. CASIA v2.0 was employed for training the proposed model and the other compared models. This dataset has 5123 tampered images totally, including 1851 splicing images and 3272 copy–move images, which are more than the other datasets used for testing. The other three datasets were used as testing datasets in order to evaluate the generalization ability of the models trained on the CASIA v2.0. The CASIA v1.0 dataset contains 921 tampered images, each sized at 384 × 256, using splicing and copy–move dataset manipulations. The Columbia Uncompressed dataset focuses on the splicing manipulation of uncompressed images with 180 tampered images. The ground truth masks of CASIA v1.0 are obtained by thresholding the difference between tampered and original images, while the ground truth masks of Columbia are provided by the producer. The DSO-1 dataset contains 100 spiced images, including indoor and outdoor images, with the resolution of each image being 2048 × 1536 pixels. These spiced images are created by adding one or more individuals to the authentic images. The ground truth masks of DSO-1 are provided by the producer.

**Table 1.** Summaries of the datasets.

| Sets | Types | Splicing | Copy–Move | Total |
|---|---|---|---|---|
| Training Set | CASIA v2.0 | 1851 | 3272 | 5123 |
| Testing Set | CASIA v1.0 | 463 | 458 | 921 |
| | Columbia | 180 | 0 | 180 |
| | DSO-1 | 100 | 0 | 100 |

In order to compare performance, we investigated and evaluated our proposed method and some other existing splicing localization algorithms, such as CNN-LSTM [8], MFCN [9], DCT [25], BLK [16], EXIF-SC [26], SpliceRadar [26], and Noiseprint [26]. Here, CNN-LSTM is a hybrid framework of CNN and long short-term memory (LSTM) cells, consisting of five convolutional layers and an LSTM network with three stacked layers. MFCN is a multi-task fully convolutional network, which uses two output branches for multi-task learning. DCT is a passive method to detect digital image forgery by measuring its quality inconsistency of blocking artifact. BLK was proposed to blindly extract the block artifact grids (BAGs) from JPEG images and check abnormal BAGs routinely. EXIF-SC, SpliceRadar, and Noiseprint are three non-end-to-end deep learning-based splice localization tools, which were proposed recently and have shown good performance in image splice detection. We computed each method's average $F_1$ and *MCC* scores on CASIA v1.0, Columbia Uncompressed and DSO-1 datasets, with the results illustrated in Table 2.

**Table 2.** The average $F_1$ scores and *MCC* metric of all methods on three test datasets.

| Methods | CASIA v1.0 | | Columbia | | DSO-1 | |
|---|---|---|---|---|---|---|
| | $F_1$ Score | *MCC* | $F_1$ Score | *MCC* | $F_1$ Score | *MCC* |
| DCT | 0.3005 | 0.2516 | 0.5199 | 0.3256 | 0.4876 | 0.5317 |
| BLK | 0.2312 | 0.1769 | 0.5234 | 0.3278 | 0.3177 | 0.4151 |
| CNN-LSTM | 0.5011 | 0.5270 | 0.4916 | **0.5074** | 0.4223 | 0.5183 |
| MFCN | 0.5182 | 0.4935 | **0.6040** | 0.4645 | 0.4810 | **0.6128** |
| EXIF-SC | 0.6195 | 0.5817 | 0.5181 | 0.4512 | **0.5285** | 0.5028 |
| SpliceRadar | 0.5946 | 0.5397 | 0.4721 | 0.4199 | 0.4727 | 0.5429 |
| Noiseprint | 0.6003 | 0.5733 | 0.5218 | 0.4255 | 0.5085 | 0.6019 |
| Our model | **0.6457** | **0.5941** | 0.5386 | 0.4278 | 0.5187 | 0.5962 |

As shown in Table 2, we find that our model outperforms all the other compared methods on CASIA v1.0 in terms of both $F_1$ score and *MCC*. Specifically, the $F_1$ score yielded by our model is 0.6457, which is 13 percent higher than the second-best model, Noiseprint, and the *MCC* generated by our model is also better than all the other compared models. Images in Columbia dataset are uncompressed, and the splicing traces can be easily discerned by the human eye. Models that emphasize local features are more suitable for this dataset. However, the $F_1$ score of the proposed model is the second best. Some predictions of tampered area are shown in Figure 3. For the DSO-1 dataset, the model with the best performance in terms of $F_1$ score is EXIF-SC (0.5285), and the performance of our model is the second best with its $F_1$ score being 0.5187. Although our model, trained on the CASIA v2.0 dataset, did not show the best performance on all three testing datasets, the overall performance is better than that of the other competitors. Moreover, the results in Table 2 also verify that the performance robustness of our model is better than that of the other compared models.

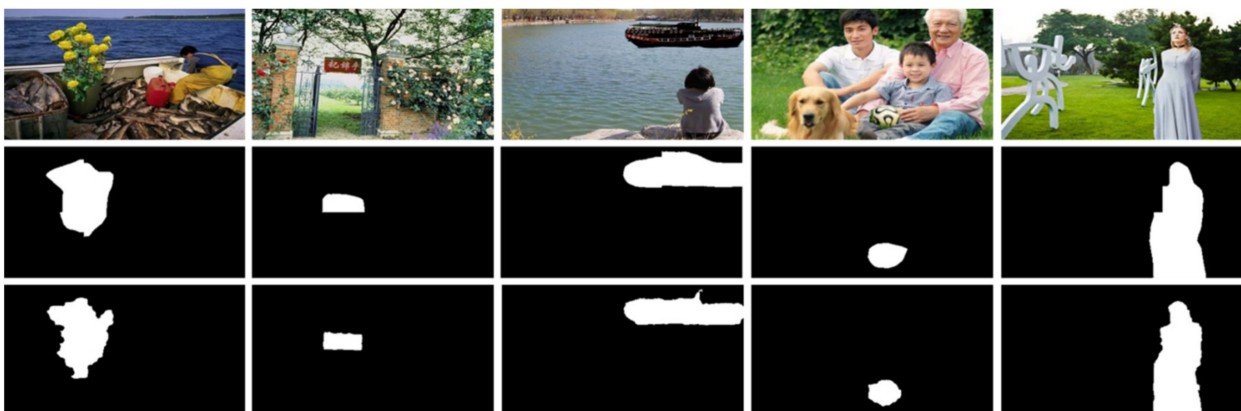

**Figure 3.** Image examples. Top row: original images; second row: binary probability maps predicted by our model; third row: the ground truth of tampered area.

We also conducted several ablation studies with various methods, including Baseline (training with seg_loss4), seg_total (training with seg_total), self-attention (training with self-attention mechanisms), and OHEM (fine-tuning with hard negative examples) on the CASIA v1.0 dataset. The results are presented in Table 3 as below.

**Table 3.** The average $F_1$ scores of proposed method on the CASIA v1.0 dataset.

| Baseline | Seg_Total | Self-Attention | OHEM | $F_1$ Score | *MCC* |
|---|---|---|---|---|---|
| √ | | | | 0.5796 | 0.5010 |
| √ | √ | | | 0.6085 | 0.5340 |
| √ | | √ | | 0.6101 | 0.5492 |
| √ | | | √ | 0.5945 | 0.5376 |
| √ | √ | √ | √ | 0.6457 | 0.5941 |

According to Table 3, we can see that the model trained with self-attention, OHEM, and multi-scale guided learning performed the best on the CASIA v1.0 dataset, indicating that multi-scale guided learning and OHEM are both helpful to detect the tampered area. The $F_1$ score of the model trained with seg_total is about 3% higher than the model trained with seg_loss4. This means that the model trained with the summation segmentation loss can activate the response of feature maps to corresponding resolution tampered regions. Moreover, by comparing the corresponding results in Table 3, we can observe that self-attention mechanisms and OHEM also contribute to the localization of tampered areas. Some output examples of the model trained with different methods are shown in Figure 4.

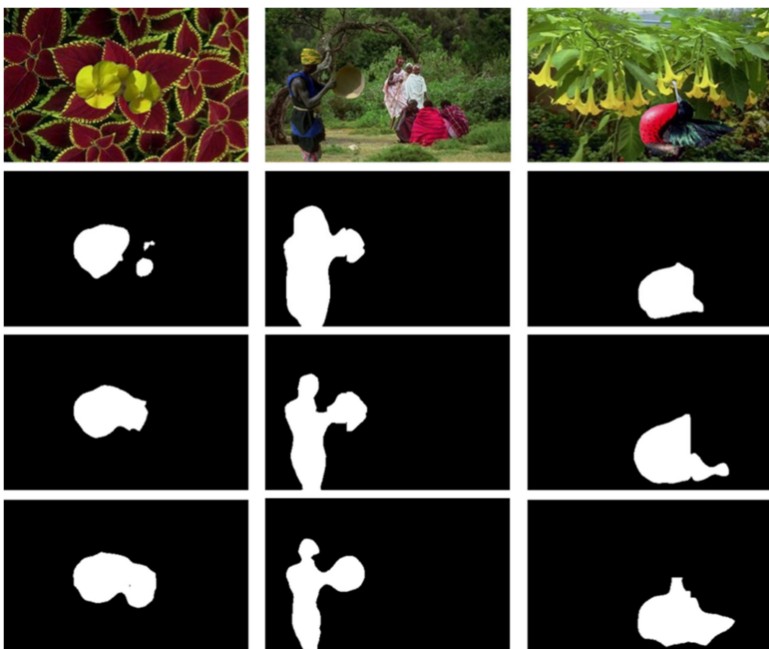

**Figure 4.** First row: tampered images; second row: the prediction of model trained with seg4; third row: the prediction of model trained with seg_total; fourth row: the prediction of model trained with seg_total, self-attention, and OHEM.

**Furthermore,** we compared the performance of our proposed model on the spliced images before and after the JPEG compression. The CASIA v1.0, Columbia Uncompressed, and DSO-1 datasets were also used for this experiment. The images are originally in JPG or TIF format, which we compressed using two different quality factors, i.e., 70 and 90. Table 4 shows the average $F_1$ scores on the original images and the JPEG compressed images by using the two different quality factors. From this table, we can see that the segmentation performance of our model has a slight degradation on the Columbia Uncompressed dataset, but a more significant one on the CASIA v1.0. The reason may be that the quality of the images in Colombian Uncompressed dataset is higher than that of the CASIA v1.0 dataset and DSO-1 dataset, and if a tampered image is generated by JPEG compression of low quality, the tampered traces of the image are concealed.

**Table 4.** The average $F_1$ scores of our model for JPEG compressed images in the three datasets.

| Datasets | $F_1$ Score | | |
| --- | --- | --- | --- |
| | Original (No Compression) | JPEG Quality = 90 | JPEG Quality = 70 |
| CASIA v1.0 | 0.6457 | 0.5355 | 0.2466 |
| Columbia | 0.5386 | 0.5341 | 0.5300 |
| DSO-1 | 0.5187 | 0.5025 | 0.4712 |

## 5. Conclusions

In this paper, we proposed a novel deep learning model for localization of tampered regions in an image. With the help of the multi-scaled learning strategy, the model can extract the global features of the image. In addition, implementation of self-attention mechanisms makes the model pay more attention to the tampered region rather than the content information of the image, and also helps the model enhance its ability to detect the tampered area. The experimental results on three well-studied benchmark datasets show that the proposed model, which were trained by the CASIA v2.0, has better overall performance than those of the other compared state-of-the-art models trained by the same training dataset. This means that our model is effective in solving the task of tampered splicing localization. In the future, we will make more attempts to improve the ability of boundary detection of the model on images with low JPEG quality compression.

**Author Contributions:** All authors contributed to the study conception and design. Conceptualization, methodology, investigation, formal analysis, and writing—original draft preparation, Z.L. and Q.Y.; writing—review and editing and supervision, J.S. All authors have read and agreed to the published version of the manuscript.

**Funding:** This research was funded in part by the National Key Research and Development Program of China (grant no: 2018YFC1603303, 2018YFC1604004), the National Science Foundation of China (grant no: 61672263) and the Key Project of Natural Science Research of Anhui Higher Education Institutions of China (KJ2021A1331, KJ2021ZD0152).

**Data Availability Statement:** CASIA datasets from https://www.kaggle.com/sophatvathana/casia-dataset (accessed on 14 April 2022); Columbia Uncompressed dataset and DSO-1 dataset from https://github.com/yizhe-ang/fake-detection-lab (accessed on 14 April 2022).

**Conflicts of Interest:** The authors declare no conflict of interest.

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
