# Peer review of "A Novel Deep Learning Architecture with Multi-Scale Guided Learning for Image Splicing Localization"

_electronics, doi:10.3390/electronics11101607_

Round 1
Reviewer 1 Report
This paper proposes a novel deep learning model for image splicing localization, which not only considers local image feature, but also take global information of images using multi-scale guided learning strategy. I recommend for publication and have the following suggestions.
- I still feel that the quality of the Abstract, Introduction and Conclusion can be improved.
- In Table 3, authors provided few F score of the model trained with a collection of parameters. It is better if authors provide all results of all possible parameters and report with both F score and MCC.
- What is the rational that authors selected only 70 and 90 in this study?
- Authors used CNN as the core predictor. But, the information of the parameter optimization was not found in the current version.
Reviewer 2 Report
In this paper, the author proposes a deep learning model for image splicing localization, which not only takes into account the local image characteristics, but also takes global information about the images to using a multi-scale guided learning strategy.
The idea of ​​using a CNN model based on the resnet-50 to extract the features is not an original idea but it has already been used by other researchers. Nevertheless, the model proposed by the author is acceptable given the results obtained.
to train the model the author used the CAISA V2.0 database and for the test the author used other CAISA V1.0 and COLUMBIA databases.
1) The author must justify this choice.
2) Why the author did not use each base separately based on the notion of cross-validation.
3) Why the author did not use the DSO-1 dataset as a test.
4) As performance measurement criteria, the author used the F1 score and the MCC without justifying his choices. the author must justify why he did not choose the Dice score or the jaccard index.
5) The author should make further comparisons with recent work published 2021 and 2022
Round 2
Reviewer 1 Report
The current version can be accepted for publication.
Reviewer 2 Report
To be accepted